# Clinical Application of Poly(ADP-ribose) Polymerase (PARP) Inhibitors in Prostate Cancer

**DOI:** 10.3390/cancers14235922

**Published:** 2022-11-30

**Authors:** Andrisha-Jade Inderjeeth, Monique Topp, Elaine Sanij, Elena Castro, Shahneen Sandhu

**Affiliations:** 1Peter MacCallum Cancer Centre, Melbourne, VIC 3065, Australia; 2Sir Peter MacCallum Department of Oncology, University of Melbourne, Melbourne, VIC 3010, Australia; 3St Vincent’s Institute of Medical Research, Fitzroy, VIC 3168, Australia; 4Department of Medicine St Vincent’s Hospital, University of Melbourne, Melbourne, VIC 3065, Australia; 5Department Medical Oncology, 12 de Octubre University Hospital, 28041 Madrid, Spain

**Keywords:** prostate cancer, PARP, PARP inhibitor, olaparib, niraparib, rucaparib, talazoparib, poly(ADP-ribose) polymerase

## Abstract

**Simple Summary:**

Stage IV (metastatic) prostate cancer remains an incurable disease despite many treatment advances in recent years. Approximately a quarter of men with advanced prostate cancer have alterations in DNA repair pathways, namely, homologous recombination repair (HRR). Poly(ADP-ribose) polymerase (PARP) is a protein directly involved in repair of damaged DNA within the cell. Therefore, inhibition of PARP leads to accumulation of DNA damage during DNA replication. Patients with underlying HRR abnormalities within cancer cells can be treated with PARP inhibitors (oral targeted therapy) to prevent cancer cell repair and progression. Our understanding of the role of PARP inhibitors as single agent and as combination therapy in prostate cancer, as well as predicting which patients will benefit most from PARP inhibitors, is evolving. In this review, we summarise the clinical trial data, as well as biomarker data, for patient selection and provide perspective as treating clinicians.

**Abstract:**

Approximately a quarter of men with metastatic castrate resistant prostate cancer (mCRPC) have alterations in homologous recombination repair (HRR). These patients exhibit enhanced sensitivity to poly(ADP-ribose) polymerase (PARP) inhibitors. Leveraging the synthetic lethality between PARP inhibition and HRR deficiency, studies have established marked clinical benefit and a survival advantage from PARP inhibitors (PARPi) in mCRPC, most notably in cancers with *BRCA1/2* alterations. The role of PARPi is evolving beyond patients with HRR alterations, with studies increasingly focused on exploiting synergistic effects from combination therapeutics. Strategies combining PARP inhibitors with androgen receptor pathway inhibitors, radiation, radioligand therapy, chemotherapy and immunotherapy demonstrate potential additional benefits in mCRPC and these approaches are rapidly moving into the metastatic hormone sensitive treatment paradigm. In this review we summarise the development and expanding role of PARPi in prostate cancer including biomarkers of response, the relationship between the androgen receptor and PARP, evidence for combination therapeutics and the future directions of PARPi in precision medicine for prostate cancer.

## 1. Introduction

Despite significant advancement in our therapeutic armamentarium in the last decade, prostate cancer remains the leading cause of cancer-related death for men worldwide [1]. Prostate cancer is highly heterogeneous, and our understanding of the different clinical and molecular subtypes and associated treatment outcomes is evolving. Approximately one-quarter of patients with metastatic castrate resistant prostate cancer (mCRPC) have alterations in homologous recombination repair (HRR), and these cancers exhibit increased sensitivity to poly(ADP-ribose) polymerase (PARP) inhibitors [2,3].

The clinical development of single agent PARP inhibitors (PARPi) in prostate cancer exploits the established paradigm of therapeutic vulnerability of HRR-deficient (HRD) cancers to PARPi [4]. Several PARPi, including two Food and Drug Administration (FDA)-approved drugs, olaparib and rucaparib, are available for use in men with mCRPC with alterations in HRR. PARPi offer the first molecularly stratified therapy for men with mCRPC and Usher in an era of personalised medicine for prostate cancer. Although the antitumour activity from PARPi is best established for prostate cancers with germline or somatic *BRCA1* and *BRCA2* alterations, clinical benefit also extends to other alterations in HRR genes, namely, *PALB2*, *BRIP*, *FANC, RAD51B, RAD51C, RAD51D, RAD54L, CHEK2,* and, to a lesser extent, *ATM* and *CDK12* [5,6,7,8,9]. Notably, the depth and durability of clinical benefit to single agent PARPi differs according to the genomic alterations and likely functional role in HRR, as well as redundancy in the DNA repair pathway [3,6,8,9]. 

Currently, the majority of PARPi efficacy data in prostate cancer are defined in the context of single HRR gene alteration, and our understanding of the implications of co-occurring genomic alterations remains limited. Prospective data suggest that *BRCA2* and co-occurring somatic alterations, including *RB1* loss and *MYC* amplification, define an aggressive subtype of prostate cancer, the therapeutic implications of which are currently not well understood [10]. Further, our understanding of the interface between tumour genomics and the immune microenvironment remains rudimentary and is the subject of ongoing research [11]. There is also a need to understand optimal treatment sequencing, as data suggest that earlier treatment with PARPi may be beneficial through modulation of androgen receptor (AR) signalling that may have downstream effects on duration of maintaining hormone responsive disease [12,13]. 

More recently, therapeutic approaches aimed at broadening the use of PARPi in prostate cancer beyond those harbouring HRR defects include combination strategies that leverage the impact of PARP inhibition on radiosensitisation, immune modulation, and enhanced DNA replication stress, all ultimately leading to enhanced cancer cell lethality (Figure 1). In this article, we review the development and increasing role of PARPi in prostate cancer as monotherapy and in combination with other therapeutics. We review the research to date on predictive biomarkers for optimal patient selection and strategies to extend the use of PARPi beyond tumours harbouring HRR defects.

## 2. PARP and PARP Inhibitors

PARP1 and 2 are members of the PARP enzyme family and are primarily implicated in repairing endogenous cellular and exogenous insults [14,15]. PARP1 and PARP2 act as key DNA damage sensors and signal transducers that mediate DNA single-strand break repair (SSBR) via base excision repair (BER). Inhibiting PARP1/2 with PARP inhibitors results in impaired BER, collapse of replication forks, and secondary accumulation of double-strand DNA breaks (DSBs) that are highly deleterious to the cell if not repaired with fidelity [4]. In the setting of DSBs, cells rely on HRR to attain high-fidelity repair that restores the DNA sequence at the site of the DNA break [16]. Cells with HRR deficiency due to loss of *BRCA1/2* function or other key proteins implicated in the HRR pathway rely on nonhomologous end joining (NHEJ) or microhomology-mediated end joining (Alt-NHEJ), a Non-conservative, error-prone repair process that can give rise to DNA alterations and genomic instability, thereby potentially fostering cancer pathogenesis and progression [17]. Beyond blocking BER, PARP inhibitors have also been shown to exert cytotoxic activity by directly trapping PARP on the DNA strand, thereby interfering with the catalytic cycle of PARP [18]. Furthermore, the accumulation of single-strand DNA replication gaps in *BRCA*-deficient cells was recently shown to underlie response to PARPi [19,20]. Two seminal papers in 2005 established the synthetic lethality between PARP inhibition and *BRCA1/2* loss in preclinical models and these data, in conjunction with the proof-of-concept phase I study of olaparib, underpin the therapeutic strategy for PARPi in patients with *BRCA1/2* mutant tumours [4,21,22]. The term “BRCAness” was coined to described non-germline *BRCA1* or *BRCA2* mutant cancers that share the phenotypic characteristics of selective therapeutic vulnerability to PARP inhibition due to deficiency of other key proteins involved in HRR, such as *RAD51, RAD54, ATM, ATR, NBS1, PALB2, and the FANC* gene family [23]. 

Beyond DNA repair, PARP1 and 2 have lesser known but multifold effects on transcriptional regulatory function including modulating AR function, apoptosis, and immune function, all of which might contribute to the anticancer activity observed in the clinic (Figure 1) [12,13,24,25,26]. The complex interplay between DNA repair, PARP, and AR, can potentially be exploited with combination therapeutic strategies to enhance DNA damage and cancer cell death, even amongst cancers without HRR alterations.

PARP is involved in multiple cellular processes, including DNA repair, transcription and replication, chromatin remodelling, and transcriptional regulation. PARP inhibitors can enact cell death by exploiting existing defective DNA repair. Alternatively, in DNA repair-proficient cells, combination strategies can be used to sensitise cells to PARP inhibition.

A. PARP inhibition results in genomic instability and cell death in DNA repair-defective cells in a process known as synthetic lethality. PARP inhibitors enact synthetic lethality in these cells in a number of ways: (i) by disrupting base excision repair, the process by which single-strand breaks (SSBs) are repaired, SSBs are then converted to double-strand breaks (DSBs) by the cells replication machinery; (ii) trapping of PARP1 at DNA damage sites, blocking access to other repair proteins; (iii) stopping the initiation of homologous recombination, the high-fidelity pathway of DSB repair, by preventing *BRCA1* recruitment by PARP; and (iv) hyperactivation of nonhomologous end joining, the more error-prone DSB repair pathway, leading to genomic instability. B. Immunomodulatory effects of PARP inhibition can be combined with checkpoint inhibitors. PARP inhibition enhances activation and recruitment of T cells by increasing PD-L1 expression and the release of neoantigens, as well as activating STING, leading to release of interferons and chemoattractants. C. Androgen receptor inhibition exerts a synergistic effect with PARP inhibitors by impairing DNA repair function. Furthermore, DNA repair genes are direct targets of androgen receptors, and androgen receptor pathway inhibition downregulates DNA repair, inducing an HRR-deficient phenotype. Furthermore, PARP1 has dual roles in DNA repair and controlling AR signalling, therefore sensitising cells to ARPI. D. Radioligand ligand therapy induces dsDNA breaks; thus, in conjunction, PARP inhibition increases cell death.

## 3. PARP Inhibitors in Prostate Cancer 

Prostate cancer pathogenesis, similar to all malignancies, is driven by the acquisition of genetic and epigenetic alterations because of impaired DNA damage-repair. In addition, there is cross-talk between AR signalling and modulation of DNA damage response via gene expression that may contribute to disease progression [27,28]. Recent data have established HRR defects are found in between 20–25% of men with mCRPC, with approximately equal rates of germline and somatic alterations [2,29]. Rates of germline alterations in localised disease are reported to be higher in high-risk disease (6%) compared to low- to intermediate-risk cancers (2%) and further enriched in patients with mCRPC (10–12%), highlighting the relationship between HRR defects with an aggressive clinical trajectory [2,3,30,31,32]. This discovery of HRR alterations in mCRPC has led to several clinical trials designed to exploit the previously established synthetic lethality of PARP inhibitors in HRR-deficient cancers [4,21,33].

## 4. Phase II Clinical Trials of PARP Inhibitors in Prostate Cancer

TOPARP was a two-part, phase II study of olaparib in men with mCRPC based on the hypothesis that a subset of unselected men with “sporadic” mCRPC would respond to PARP inhibition due to underlying HRR defects [3,6]. At the time of study initiation, there were limited data on the presence of HRR defects in prostate cancer and no established predictive biomarkers for PARPi response beyond germline *BRCA* mutations. Therefore, an adaptive design was employed to co-develop the PARPi, olaparib, and the predictive biomarkers of response. TOPARP-A enrolled 49 molecularly unselected, heavily pre-treated mCRPC patients with the interrogation of putative predicative biomarkers embedded as a key design feature of the study [3]. The primary endpoint was composite response rate defined as either an objective radiological response (ORR) by response evaluation criteria in solid tumours 1.1 (RECIST 1.1), a decline in prostate specific antigen (PSA) level of at least 50% (PSA_50_), and/or conversion in circulating tumour cell (CTC) count from ≥5 cells/7.5 mL of blood to <5 cells/7.5 mL of blood, confirmed in a second reading 4 or more weeks apart [3,6]. Notably, 16 out of the 49 patients (33%; 95% confidence interval (CI); 20–48) achieved a composite response [3]. Whole-exome sequencing performed on tumour samples revealed that 33% of patients had deleterious biallelic alterations in HRR genes, with 14/16 (88%) of these patients demonstrating a response to olaparib, compared with only 6% of the biomarker negative cohort [3]. This was the first study to report responses in mCRPC patients with both germline and somatic alterations in a *BRCA 1/2* cohort, as well as additional subgroups with alterations in *ATM, CHEK2, PALB2,* and other genes implicated in the HRR pathway [3]. Of the seven patients with biallelic *BRCA2* alterations, two harboured somatic homozygous deletions, two had heterozygous somatic alterations, and three patients had germline mutations with co-occurring somatic loss. The five patients with *ATM* alterations who responded consisted of two somatic and three germline mutations. Two out of three with germline *ATM* mutations had co-occurring somatic alterations (biallelic). Radiologic progression-free survival (rPFS) and median overall survival (OS) were longer in the biomarker-positive group compared with biomarker-negative patients, 9.8 versus (vs.) 2.7 months (*p* < 0.001) and 13.8 vs. 7.5 months (*p* = 0.05), respectively [3]. 

Based on the promising responses identified in Part A, a validation cohort in Part B (TOPARP-B) was initiated with mandatory preselection for biallelic alterations in HRR genes [3,6]. TOPARP-B randomised 98 patients to receive either 400 mg or 300 mg olaparib twice daily [6]. The composite response was 54.3% (25 out of 46; 95% CI 39.0–69.1) in evaluable patients on the 400 mg olaparib cohort, and 39.1% (18/46; 95% CI 25.1–54.6) in the 300 mg cohort [6]. The ORR in patients with RECIST1.1 measurable disease was 24.2% (8/33) and 16.2% (6/37) for the 400 mg and 300 mg olaparib cohorts, respectively [6]. Response rates and duration of response differed according to gene aberrations: the composite response was 83.3% (25/30; 95% CI 65.3–94.4) and rPFS was 8.3 months (95% CI 5.5–13.0) for *BRCA1/2*; 57% (4/7; 95% CI 18.4–90.1) and rPFS 5.3 months (95% CI 0.4: not estimable) for *PALB2*; 36.8% (7/19; 95% CI 16.3–61.6) and rPFS 5.8 months (95% CI 4.4–10.9) for *ATM*; 25% (5/20; 95% CI 8.7–49.1) and rPFS 2.9 months (95% CI 2.6–7.5) for *CDK12*; and in other gene alterations, the composite response was 20% (4/20; 95% CI 5.7–43.7) with rPFS of 2.8 months (95% CI 2.6–4.3) [6]. 

Building on the TOPARP data, several phase II and III studies examining different PARPi in mCRPC were initiated (Table 1). The TRITON2 phase II study of rucaparib enrolled 192 mCRPC patients with disease progression after one or two lines of androgen receptor pathway inhibitors (ARPIs), as well as prior taxane chemotherapy with deleterious germline or somatic alterations in HRR genes, to a *BRCA1/2* (*n* = 115) and non-*BRCA* (*n* = 78) cohort of 13 other genes, including *ATM, CDK12, CHEK2, FANCA, PALB2,* and *RAD51,* using tissue or circulating tumour DNA assay [8,9,34]. Biallelic alterations in HRR genes was not a prerequisite for trial entry. The primary endpoint was ORR (RECIST 1.1), with PSA decline of ≥50% (PSA_50_ response) as a secondary endpoint [8,9,34]. The *BRCA* cohort predominantly consisted of patients with *BRCA2* alterations (102/115; 89%) with germline *BRCA* alterations identified in 41% (47/115). The ORR in the *BRCA* cohort was 43.5% (27/62; 95% CI 31.0–56.7%) with a PSA_50_ response rate of 54.8% (63/115; 95% CI 45.2–64.1%) [9,34]. Patients with homozygous *BRCA1/2* loss demonstrated a PSA_50_ response rate of 80% (12/15; 95% CI 51.9–95.7) and ORR 70% (7/10; 95% CI 34.8–93.3) compared with PSA response 50% (11/22; 95% CI 28.2–71.8) and ORR 40% (4/10; 95% CI 12.2–73.8) in patients with heterozygous *BRCA1/2* alterations [34]. The ORR for patients with germline *BRCA*1/2 alterations was 48.5% (11/24; 95% CI 25.6–67.2) compared with 42.1% (16/38; 95% CI 26.3–59.2) in patients with *BRCA1/2* somatic alterations [9,34]. In the non-*BRCA* cohort, the ORR by gene alterations included *ATM* 10.5% (2/19), *CDK12* 0% (0/10), and *CHEK2* 11.1% (1/9) in addition to 28.6% (4/14) responses also seen in other non-*BRCA* HRR alterations, including *PALB2, FANCA, RAD51B,* and other subgroups [8]. 

The GALAHAD phase II study of niraparib enrolled 289 patients with mCRPC, of whom most had received three or more lines of therapy and needed to have documented progression on ARPI and prior taxane chemotherapy [5]. Patients were selected for biallelic alterations in HRR related genes across two cohorts: *BRCA 1/2* alterations (cohort A, *n* = 142) and non-*BRCA* alterations in other HRR genes including *ATM, CHEK 2, FANCA, PALB2, HDAC2,* and *BRIP1* (cohort B, *n* = 81) [5]. The primary endpoint, ORR (in patients with RECIST1.1 measurable disease), was 34.2% (26/76; 95% CI 23.7–46.0) for the *BRCA 1/2* cohort and 10.6% (5/47; 95% CI 3.5–23.1) in the non-*BRCA* cohort [5]. The composite response rate (defined as either an OR by RECIST 1.1, CTC conversion to <5/7.5 ml blood, or PSA_50_) was 58% (82/142; 95% CI 49.2–66.0) for the *BRCA* cohort and 15% (12/81; 95% CI 7.9–24.5) for the non-*BRCA* cohort. With a median follow-up of 10 months, the median OS for the *BRCA* cohort and non-*BRCA* cohorts were 13.01 (11.04–14.29) and 9.62 (8.05–13.44) months, respectively [5]. 

The TALAPRO-1 phase II study of talazoparib enrolled 128 patients with mCRPC who had prior taxane chemotherapy and ARPI with pathogenic germline or somatic alterations in any of 11 prespecified DNA repair genes: *ATM, ATR, BRCA1, BRCA2*, *CHEK2*, *FANCA*, *MLH1, MRE11A*, *NBN*, *PALB2*, and *RAD51C* [35]. Overall, the primary endpoint, ORR, in patients with RECIST 1.1 measurable disease was 29.8% (31/104, 95% CI 21.2–39.6) [35]. Across the different cohorts, the ORR was 46% (28/61) for *BRCA1/2* 25% (1/4) for *PALB2*, 12% (2/17) for *ATM,* and 0% for (0/22) other gene alterations [35]. Responses were independent of germline or somatic DNA repair gene alterations; however, a PSA decline of 30% or greater was predominantly seen in patients with homozygous alterations in HRR genes [35]. 

Collectively, these phase II trials evaluating the efficacy of different PARP inhibitors were largely focused on heavily pre-treated men with mCRPC but differed in terms of methodology of HRR gene testing and number of lines of prior treatment exposure. TOPARP and GALAHAD required patients with biallelic alterations in HRR genes, whilst TRITON2 and TALAPRO-1 enrolled patients with either germline or somatic alterations, and monoallelic or biallelic loss was not ascertained for enrolment but was retrospectively analysed. Across the studies, the ORR and PSA_50_ response rates differed, likely due to the significant heterogeneity among patients and differing proportions and types of HRR alterations, including zygosity status and number of lines of therapy, as well as other prognostic clinical features (see Table 1).

Several publications have attempted to compare different PARPi across trials based on different therapeutic responses, toxicity profiles, and potential surrogate markers of efficacy [36]. Variations between PARPi include differing abilities to inhibit enzymatic activity, PARP-trapping ability, and drug distribution within tissue; however, the clinical implications of these measures on both efficacy and toxicity is less well defined, with most PARPi showing an ORR in the range of 34.2–52% in *BRCA1/2* altered prostate cancer [3,5,6,7,34,35,36,37,38,39]. Overall, PARPi are well tolerated with mild gastrointestinal and haematological class effect toxicities that are largely manageable with dose interruptions, treatment delays, dose reductions, and supportive measures [40].

**Table 1 cancers-14-05922-t001:** Single agent PARP inhibitor trials in metastatic castrate resistant prostate cancer.

Trial Name	Phase of Trial (n = Number of Patients)	Treatment Arms	Patient Population	HRR Biomarker Inclusion	Key Endpoints	Key Efficacy Results	Responses in Molecular Subtypes
Phase II PARP inhibitor single agent trials in metastatic castrate resistant prostate cancer after taxane therapy							
TOPARP-A [3]	Phase II (n = 50)	Olaparib 400 mg BD.	mCRPC, progressed after 1 or 2 taxane-based regimes.	Unselected; subsequent molecular characterization for HRR alterations.	PE: Composite response rate; radiological, PSA_50_ or CTC reduction. SE: rPFS, OS.	All patients: Composite response 33% (95% CI 20–48), PSA_50_ 22%.	88% composite response in HRR altered (BM+) cohort vs. 6% without HRR alterations (BM-) rPFS 9.8 (BM+) vs. 2.7 mo (BM-) (*p* < 0.001). Median OS 13.8 (BM+) vs. 7.5 mo (BM-) (*p* = 0.05). Composite response BRCA2: 100%, ATM: 80%.
TOPARP-B [6]	Phase II (n = 98)	Olaparib 300 mg BD or 400 mg BD.	mCRPC progressed after 1 or 2 taxane-based regimes.	Biallelic deleterious HRR alterations.	PE: Composite response rate (defined as TOPARP-A) SE: rPFS, OS.	Composite response 400 mg: 54.3% (95% CI 39.0–69.1), 300 mg: 39.1% (25.1–54.6). Median rPFS 5.5 mo (95% CI 4.4–8.3) 400 mg vs. 5.6 mo (3.7–7.7) 300 mg. Median OS 14.3 mo (9.7–18.9) 400 mg vs. 10.1 mo (9.0–17.7) 300 mg.	Composite response BRCA1/2 83.3% (65.3–94.4), PALB2 57.1% (18.4–90.1), ATM 36.8% (16.3–61.6), CDK12 25% (8.7–49.1), Other 20% (5.7–43.7).
TRITON 2 [8,9,34]	Phase II (n = 115 BRCA 78 non-BRCA)	Rucaparib 600 mg BD.	mCRPC, progressed after at least 1 taxane-based regimen and 1 ARPI.	Bi- or monoallelic, germline, or somatic deleterious BRCA1/2 mutation or other prespecified HRR gene.	PE: ORR SE: DOR, PSA_50_.	BRCA1/2 ORR 43.5% (95% CI 31.0–56.7), PSA_50_ 54.8% (45.2–64.1). Median DOR not reached (NR; 95% CI 6.4 mo-NR), PSA_50_ 54.8% (95% CI 45.2–64.1).	PSA_50_ gBRCA 61.7% (95% CI 46.4–75.5), sBRCA 50% (37.6–62.4). ORR gBRCA 45.8% (95% CI 25.6–67.2), sBRCA 42.1% (26.3–59.2). ATM: ORR 10.5%, PSA_50_ 4.1%. CDK12: ORR 0%, PSA_50_ 6.7%. CHEK12: ORR 11.1%, PSA_50_ 16.7%.
GALAHAD [5]	Phase II (n = 289)	Niraparib 300 mg OD.	mCRPC progressed after at least 1 taxane-based regimen and 1 ARPI.	Biallelic HRR or germline pathogenic BRCA1/2 alterations (BRCA cohort) or biallelic alterations in other prespecified DDR genes (non-BRCA cohort)—included ATM, BRIP1, CHEK2, FANCA, HDAC2, PALB2.	PE: ORR (trial amended PE) in BRCA cohort. SE: ORR in non-BRCA, OS, rPFS, composite response (radiological, PSA_50_ or CTC reduction).	ORR BRCA: 34.2% (95% CI 23.7–46.0). Non-BRCA: 10.6% (3.5–23.1). OS BRCA 13.01, non-BRCA 9.63 mo. rPFS BRCA 8.08, non-BRCA 3.71 mo. Composite response BRCA 58%, non-BRCA 15%.	Results for BRCA1 and BRCA2 cohorts reported together.
TALAPRO 1 [35]	Phase II (n = 128)	Talazoparib 1 mg OD.	mCRPC progressed after at least 1 taxane-based regimen and 1 ARPI.	Mono- or biallelic HRR alterations (CDK12 excluded).	PE: ORR, SE: OS rPFS, PSA_50_.	ORR 29.8% (95% CI 21.2–39.6). Median OS 16.4 mo (95% CI 12.2–19.9). Median rPFS 5.6 mo (95% CI 3.7–8.8). PSA_50_ 42% _._	ORR: BRCA1/2 46%, BRCA2 46%, PALB2 25%, ATM 12%, Other 9%.
Phase III PARP inhibitor single agent trials in metastatic castrate resistant prostate cancer							
PROfound [7]	Phase III (n = 387)	Olaparib 300 mg BD vs. physicians choice of enzalutamide or abiraterone.	mCRPC after at least 1 ARPI (previous taxane was allowed).	Bi- or monoallelic, somatic, or germline, deleterious HRR alterations. Cohort A: BRCA1/2 or ATM mutations. Cohort B: other 12 HRR genes mutations.	PE: rPFS in cohort A. SE: ORR, PSA_50_, OS.	Median rPFS Cohort A + B 5.8 olaparib vs. 3.5 mo control (HR 0.49 95% CI 0.38–0.63). ORR 22% olaparib vs. 4% control (OR 5.93; 95% CI 2.01–25.40). Median OS (prelim) 17.5 olaparib vs. 14.3 mo control (HR 0.67; 95% CI 0.49–0.93). PSA_50_ 30% olaparib vs. 10% control.	Median PFS Cohort A: 7.4 olaparib vs. 3.6 mo control (HR 0.34; 95% CI 0.25–0.47). Cohort B: 5.8 mo vs. 3.5 mo (HR 0.49, 95% CI: 0.38–0.63). Cohort A: ORR 33% olaparib vs. 2% control (OR 20.86; 95% CI 4.18–379.18). Cohort A: PSA_50_ 43% olaparib vs. 8% control. Cohort A: Median OS (prelim) 18.5 olaparib vs. 15.1 mo control (HR 0.64; 95% CI 0.43–0.97).
TRITON3 [41]	Phase III (n = 405)	Rucaparib vs. physicians choice of either enzalutamide, abiraterone or docetaxel.	mCRPC progressed after ARPI (no prior chemotherapy in castrate resistant setting).	BRCA1/2 or ATM alteration.	PE: rPFS. SE: OS, ORR.	Prelim: ITT population (ATM and BRCA) Median rPFS 10.2 rucaparib vs. 6.4 mo control:HR 0.50 (95% CI 0.36–0.69). OS data immature.	Prelim: BRCA: Median rPFS 11.2 rucaparib vs. 6.4 mo control (*p* < 0.0001). ATM: Median rPFS 8.1 rucaparib vs. 6.8 mo control (*p* = 0.8421). OS data immature.

BD: twice daily; OD: daily; ARPI: androgen receptor pathway inhibitors; HRR: homologous recombination repair; PE: primary endpoint(s); SE: secondary endpoint(s); ORR: overall response rate; PSA_50_: reduction in PSA level of ≥50%; CTC: circulating tumour cell response; rPFS: radiological progression-free survival; mo: months; HR: hazard ratio; BM+: biomarker-positive, HRR alterations; BM-: biomarker-negative, no HRR alterations; *gBRCA:* germline *BRCA* alteration; *sBRCA:* somatic *BRCA* alteration; OR: odds ratio; prelim: preliminary result; ITT: intention to treat population.

## 5. Phase III Clinical Trials of PARP Inhibitors in Prostate Cancer

The phase III PROfound trial evaluated the efficacy of olaparib in men with mCRPC and HRR alterations who had progressed after prior ARPI. Patients with *BRCA1/2* and *ATM* alterations were enrolled in cohort A (*n* = 245), while patients with 12 other prespecified HRR gene alterations were enrolled in cohort B (*n* = 142). Patients were randomised 2:1 to olaparib or physician’s choice of enzalutamide or abiraterone, depending on prior ARPI exposure [7]. The primary endpoint, median rPFS, in cohort A was significantly improved with olaparib at 7.4 months compared to 3.6 months with ARPI retreatment (HR 0.34; 95% CI 0.25–0.47; *p* < 0.001) [7]. The median rPFS was also improved with the use of olaparib vs. ARPI in the combined cohorts A and B at 5.8 vs. 3.5 months (HR 0.49; 95% CI 0.38–0.63; *p* < 0.001). The superior rPFS benefit observed in cohort A was largely driven by patients with *BRCA1/*2 alterations that made up two-thirds of the cohort. The median rPFS in the *BRCA1/2* cohort was 9.8 vs. 3.0 months (HR 0.22, 95% CI 0.15–0.32) for olaparib vs. ARPI. In contrast, there was no rPFS benefit in men with *ATM* alterations (HR 1.04; 0.61–1.87), consistent with other studies demonstrating modest PARPi antitumour activity in men with *ATM* alterations [7,42]. The confirmed ORR of 33.3% (28/84) vs. 2.3% (1/43) (odds ratio 20.9, 95% CI 4.2–379.2; *p* < 0.0001) and other key secondary endpoints including time to pain progression (HR 0.44; 95% CI, 0.22 to 0.91; *p*= 0.02) and health-related quality of life was significantly improved in patients who received olaparib compared with ARPI [7]. 

The median OS was significantly improved with olaparib compared to ARPI in cohort A at 19.1 vs. 14.7 months (HR 0.69; 95% CI 0.50 to 0.97; *p* = 0.02), in cohort B at 14.1 vs. 11.5 months (HR 0.96; 95% CI 0.63–1.49), and in the overall cohorts A and B combined at 17.3 vs. 14.0 months (HR 0.79; 95% CI 0.61–1.03) [43]. Notably, the survival advantage was observed despite 66% (83/131) patients in the control group crossing over to receive olaparib, suggesting that there is benefit from treatment with PARPi earlier in the disease course [43]. These positive data resulted in approval of olaparib in multiple jurisdictions, including European Medicines Agency (EMA), FDA, and others. Given the strength of evidence for responses from PARPi in *BRCA1/2* alterations, olaparib has consistently been approved for this cohort of patients; however, country-specific approvals across the other HRR gene alterations have varied based on interpretation of the strength of the data.

TRITON3 (NCT02975934) is a phase III randomised study evaluating rucaparib vs. physician’s choice of either enzalutamide, abiraterone, or docetaxel in 405 patients with mCRPC and an underlying HRR (*BRCA1/2* or *ATM*) alteration [41]. Preliminary results revealed that the primary endpoint of rPFS was 10.2 months for rucaparib vs. 6.4 months for physician’s choice (HR 0.61; 95% CI 0.47–0.80) [44]. In the *BRCA*-altered cohort, the rPFS was 11.2 months vs. 6.4 months in favour of rucaparib (HR 0.5; 95% CI 0.36–0.69). Notably, the rPFS difference was small in patients with *ATM* alterations (8.1 months for rucaparib vs. 6.8 months for physicians’ choice, HR 0.97; 95% CI 0.59–1.52; *p* = 0.8421) [44]. Based on the promising phase II/III data, rucaparib has received accelerated FDA approval for patients with mCRPC with *BRCA1*/*2* alterations, having progressed on both prior ARPI and taxane chemotherapy. 

## 6. Targeting Androgen Receptor (AR) and PARP Concurrently

AR blockade with ARPI results in downregulation of genes involved in DNA repair, enhanced DNA damage, and confers increased sensitivity to PARPi [12,13,28]. Separate from its well-described role in DNA repair, PARP1 is implicated in diverse transcriptional regulatory function, including modulating AR function. Preclinical studies of prostate cancer demonstrate that PARP1 enzymatic activity is enhanced and required for AR function, tumour cell growth, and progression to castration resistance [24]. The dual roles of PARP in DNA damage response and sustaining AR function taken in conjunction with data showing that abrogating AR signalling induces an “HRR-deficient” phenotype provide the basis for co-targeting AR and PARP concurrently to slow down tumour proliferation and disease progression [24]. Preclinical studies of the combination of PARP inhibitors with APRI have demonstrated improved antitumour activity compared with the respective single agent activity [13].

A phase II study (NCT01972217) evaluated the efficacy of combined treatment with olaparib and abiraterone in 142 patients with mCRPC without prior selection for HRR alterations. This study demonstrated improvement in median rPFS for the olaparib and abiraterone combination compared with abiraterone alone (13.8 months vs. 8.2 months; HR 0.65, 95% CI 0.44–0.97; *p* = 0.034) [45]. Whilst not powered for subgroup analysis nor evaluable for all patients, HRR alterations were identified in 56% (38/68) of prostate tumour samples analysed, with similar overall HRR alteration rates in the combination therapy (15%; 11/71) and monotherapy (14%; 10/71) cohorts. 

Several phase III studies are evaluating the combination of PARP and ARPIs in metastatic-hormone-resistant and -sensitive prostate cancer. The PROpel phase III study enrolled 796 patients, irrespective of HRR status, to either olaparib and abiraterone with prednisolone vs. abiraterone with prednisolone and placebo as first-line treatment for mCRPC [46]. Prior treatment with docetaxel in the hormone-sensitive setting was permitted; however, prior abiraterone exposure was an exclusion. Biomarker analysis revealed that 29% of the enrolled patients had an underlying HRR alteration, including approximately 10% with *BRCA1/2* alterations, and this was balanced across the arms. The olaparib and abiraterone combination prolonged rPFS regardless of HRR status (25 vs. 16 months; HR 0.67; 95% CI 0.56–0.81; *p* < 0.001), although the benefit was greatest in the cohort with HRR alterations (median of 13.9 m vs. not reached (NR); HR 0.50; 95% CI 0.34–0.73) compared to those without HRR alteration (24 vs. 19 months; HR 0.76; 95% CI 0.60–0.97) [47]. The combination treatment benefit was most apparent in patients with *BRCA1/2* alterations with median rPFS that was NR vs. 8.4 months; HR 0.23; 95% CI 0.12–0.43) [47]. Secondary endpoints of time to first subsequent therapy and progression-free survival 2 (PFS2) also favoured the combination arm (HR 0.76 and 0.71, respectively) [48]. Survival data remain immature, with the trend favouring the olaparib and abiraterone with prednisolone combination (HR 0.86; 95% CI 0.66–1.12; *p* = 0.29) [47].

The phase III MAGNITUDE study assessed the combination of niraparib and abiraterone with prednisolone vs. abiraterone with prednisolone and placebo in 670 patients with mCRPC who were assigned to a prespecified HRR biomarker-positive (HRR+, *n* = 423) or -negative (HRR−, *n* = 247) cohort based on pathogenic HRR-associated gene alterations (*BRCA1, BRCA2, CDK12, FANCA, PALB2, CHEK2, BRIP1, HDAC2, ATM*). Patients were permitted to have docetaxel for hormone-sensitive disease and up to four months of abiraterone for mCRPC prior to randomisation [48]. After a median follow-up of 18.6 months in the HRR+ cohort, patients with *BRCA1/2* alterations treated with niraparib and abiraterone with prednisolone demonstrated significantly improved rPFS compared to abiraterone with prednisolone (16.6 vs. 10.9 months; HR 0.53; 95% CI 0.36–0.79; *p* = 0.001). Across the entire HRR+ cohort, rPFS was improved in the niraparib combination arm (16.5 vs. 13.7 months; HR 0.73; 95% CI 0.56–0.96; *p* = 0.022) [48]. Combination therapy also delayed time to cytotoxic chemotherapy (HR 0.59; 95% CI 0.39–0.89; *p* = 0.011), as well as time to symptomatic progression (HR 0.69; 95% CI 0.47–0.99; *p* = 0.04), compared to abiraterone and placebo in the HRR+ cohort [48]. A prespecified futility analysis using a composite efficacy endpoint of PSA and radiological progression was performed in the HRR- cohort (*n* = 233) showing no benefit from the combination intervention (HR 1.09; 95% CI 0.75–1.57; *p* = 0.66) [48].

TALAPRO-2 (NCT03395197) is a phase III study evaluating the efficacy of talazoparib and enzalutamide vs. enzalutamide in 1:1 randomisation as first-line therapy for mCRPC with the primary endpoint of rPFS. The study enrolled 1018 patients across two cohorts: cohort 1 included patients unselected for HRR status, and cohort 2 were prespecified for HRR alterations, including *BRCA1, BRCA2, PALB2, ATM, ATR, CHEK2, FANCA, RAD51C, NBN, MLH1, MRE11A,* and *CDK12* [49]. A recent media release revealed that the combination of talazoparib and enzalutamide met the trial’s primary endpoint and improved rPFS compared to placebo and enzalutamide, exceeding the prespecified HR 0.696 [50].

CASPAR (NCT04455750) is a phase III study of enzalutamide and rucaparib vs. enzalutamide, enrolling 984 men with mCRPC unselected for HRR alterations to assess the co-primary endpoints rPFS and OS [51]. 

Several phase III trials are evaluating the combination of PARPi and ARPI for newly diagnosed metastatic hormone-sensitive prostate cancer (mHSPC). TALAPRO-3 (NCT04821622) will enrol 550 patients to assess the combination of talazoparib and enzalutamide vs. enzalutamide in mHSPC with HRR alterations, including *ATM, ATR, BRCA1/2, CDK12, CHEK2, FANCA, MLH1, MRE11A, NBN, PALB2,* and *RAD51C* [52].

Amplitude (NCT04497844) will randomise 788 patients with mHSPC and underlying alterations in *BRAC1/2, BRIP1, CDK12, CHEK2, FANCA, PALB2, RAD51B*, and *RAD54L* to treatment with niraparib and abiraterone vs. abiraterone and placebo [53].

## 7. Biomarkers of Response and Resistance

The recent approval of PARPi in patients with HRR alterations in multiple jurisdictions has ushered in routine genomic testing and the first molecularly stratified therapy in mCRPC. Multiplexed genomic testing offers the opportunity to optimise patient selection for PARPi; however, our understanding of predictors of response is still evolving. For example, the clinical significance of *BRCA* alterations in terms of conferring sensitivity to PARPi is well established in prostate cancer; however, even within this subset of cancers, the clinical relevance of molecular features, such as germline vs. somatic origin, mutation vs. deletion, zygosity status, other co-occurring molecular aberrations, and the interface between genomics and the tumour microenvironment, is still emerging. Furthermore, our understanding of the clinical relevance of other, rarer genomic alterations directly or indirectly involved with the HRR pathway is cursory at the present time due to the limited datasets and heterogenous selection criteria across studies. Further in-depth understanding of predictive biomarkers of response is needed to improve patient stratification and to define optimal treatment selection. 

In a pooled analysis of data from the TOPARP, PROfound, TRITON2, and TALAPRO-1 trial populations with mCRPC receiving PARPi monotherapy, efficacy data were compared for *BRCA1* and *BRCA2* alterations separately [54]. The ORR was superior for *BRCA2* at 50.0% (79/158; 95% CI 42.2–57.8) compared to 26.3% (5/19; 95% CI 5.1–47.5) for *BRCA1*. Similarly, the rPFS was longer for *BRCA2* at 10.1 months (95% CI 8.9–11.6) compared to 4.1 months (95% CI 1.0–16.8) for *BRCA1.* However, the small numbers of patients with *BRCA1* alterations makes robust conclusions from these data not possible [54]. These differences in PARPi efficacy in *BRCA2-* and *BRCA1*-altered prostate cancers may relate to higher rates of germline and biallelic alterations and the potential influence from other genomic co-alterations [54]. Although commonly grouped together in most datasets, *BRCA1* and *BRCA2* have several differentiating roles in cell function as well as different implications for cancer risk and treatment response across tumour types. *BRCA2* is primarily involved in HRR repair, whilst *BRCA1* has several other cellular roles [55]. 

TRITON2 examined responses to rucaparib in *BRCA1/2* based on germline vs. somatic alteration, zygosity status, and type of gene alteration [9]. ORRs were similar at 45.8% (11/24; 95% CI 25.6–67.2) for germline vs. 42.1% (16/38; 95% CI 26.3–59.2) for somatic *BRCA1/2* alterations [34]. TALAPRO-1 and TOPARP-B also demonstrated consistent data, showing no significant difference in ORR based on germline or somatic origin of *BRCA* alterations [35,56]. 

In TRITON2, PSA responses were reported to be higher in patients with biallelic *BRCA1/2* alterations at 75.0% (27/36; 95% CI 57.8–87.9) compared to monoallelic *BRCA1/2* alterations at 11.1% (1/9; 95% CI 0.3–48.2), noting that the BRCA cohort were predominantly made up of *BRCA2* patients (89%; 102/115). Additionally, homozygous *BRCA1/2* loss was associated with superior PSA response, 81.0% (17/21; 95% CI 58.1–94.6), compared to other alteration types, 48.9% (46/94; 95% CI 38.5–59.5) [9]. Similarly, exploratory analysis in TALAPRO-1 demonstrated numerically superior ORR 50.0% (9/18) in patients with homozygous *BRCA2* alterations vs. 44.0% (4/9) in patients with heterozygous alterations [35]. In TOPARP-B, patients with tumours harbouring homozygous *BRCA2* deletions demonstrated a superior median rPFS of 16.4 months vs. 5.6 months for germline *BRCA1/2* mutations and 8.2 months for *BRCA1/2* somatic mutations [56]. Patients with *BRCA1/2* biallelic loss had a median rPFS and OS of 9.7 and 18.9 months respectively, compared to 5.6 and 14.6 months for patients without biallelic *BRCA1/2* loss [56].

Monoallelic *BRCA1/2* alterations may lead to incomplete inactivation of HRR function, whilst biallelic deletions are likely less permissive to emergence of reversion mutations and PARPi resistance, thus possibly explaining the superior outcomes seen in these patients [9,37,54,56]. Studies specific to prostate cancer have described *BRCA2* mutation reversion after PARPi treatment, resulting in HRR restoration as a putative mechanism of treatment resistance [57,58]. 

Of the varying HRR genes, patients with *BRCA1/2* alterations have consistently demonstrated superior objective responses to PARPi (34–52%) compared to non-*BRCA1/2* (10–12%) [5,6,7,34,35,59]. *ATM* alterations show consistently lower responses, with radiological response rates ranging from 8–12%, PSA response rates of 4–13%, and composite response rates of 24–37% [60]. The authors of TRITON2 reported a lack of response to rucaparib in the cohort with *CDK12* alterations, with 0/10 patients achieving a radiographic response and only one patient (6.7%) demonstrating a short-lived PSA response [8]. In the cohort with *CHEK2* alterations, only 1/9 (11.1%) evaluable patients demonstrated a radiographic response, and 2/12 (16.7%) had a PSA response [8]. 

It would be logical to theorise that prostate cancers with *PALB2* (partner and localiser of *BRCA2*) alterations would respond to PARPi similarly to *BRCA2,* given its role in HRR. However, *PALB2* alterations have only been identified in a very small subset of patients in the recent trials, making definitive conclusions challenging. In TOPARP-B, seven patients with *PALB2* mutations were enrolled, with only four out of the seven patients identified to have biallelic loss responding to olaparib [56]. 

Overall, very limited PARPi activity has been observed for patients with *ATM* and *CDK12* alterations [6,7,8,35]. Other genes, such as *RAD51,* have demonstrated efficacy from PARPi in breast and ovarian cancer due to association with the HRR-deficient phenotype; however, the rarity of these alterations and others, such as *RAD50, FANCA, FANCM*, and *BRIP1,* makes accurate ascertainment of likely response difficult as there are few published efficacy data available [59]. 

Our understanding of the presence and implications of co-alterations such as *p53*, *PTEN,* and *MYC* are currently rudimentary, with combination alterations underrepresented in published data. PROREPAIR-A was a prospective multicentre case–control study of 73 germline *BRCA2* carriers and 127 matched non-carriers to assess the prognostic implications of *BRCA2* and other co-occurring alterations. Germline *BRCA2* carriers demonstrated more somatic co-alterations (*p* < 0.001), including *RB1* loss (40.8% vs. 11.8%, *p* < 0.01) and *MYC* amplification (51.4% vs. 22.8%, *p* < 0.01), than noncarriers [60]. Median cause-specific survival (CSS) was worse in *BRCA2* carriers, with confirmed worse independent prognostic value in somatic co-deletions in *RB1* (HR 4.13; *p* = 0.004) and *MYC* amplifications (HR 2.27; *p* = 0.033) [61]. 

Improved understanding of HRR defects beyond *BRCA1/2* mutations is still required to allow more accurate patient selection. *RAD51* foci analysis and other functional assays of “genomic scars” associated with DNA damage may identify an HRR-deficient phenotype [56,61]. In an exploratory biomarker analysis in TOPARP-B, a *RAD51* foci score as a surrogate measure of HRR function was able to identify all *BRCA1/2*-altered and *PALB-2* biallelic loss tumours in keeping with HRR loss of function [56]. Low *RAD51* foci scores were associated with longer median rPFS (9.3 vs. 2.9 months) and OS (17.4 vs. 9.5 months) compared to high *RAD51* scores amongst olaparib sensitive tumours; however, low *RAD51* scores did not capture all potential responders [56]. These data suggested that the RAD51 assay may be useful as an additional predictor of response to PARPi therapy; however, requires prospective clinical validation [56].

## 8. BRCA and HRR Testing in Advanced Prostate Cancer

The incidence of *BRCA1/2* alterations in prostate cancer are evenly split across somatic and germline alterations. For this reason, testing the tumour is currently recommended as the gold standard to identify both somatic and germline alterations in HRR genes [62]. Given the high incidence of germline alteration (~12%) and important implications for family members who may be at risk of hereditary cancers, a positive *BRCA1/2* tumour test should be followed up with germline testing and a referral to the genetics counsellors as needed [63]. In a cohort study of matched same-patient samples, tissue profiling from primary prostate at diagnosis (*n* = 470) and mCRPC biopsies (*n* = 61) demonstrated that results for HRR alterations were similar between primary and metastatic sites, highlighting that these events occur early and are largely detectable in archival material [64]. The PROfound study, the largest dataset to date, provides practical insights into testing for HRR alterations. In this study, 4858 tumour samples were sequenced, showing a marginally higher rate of NGS result (63.9%) from metastatic samples compared with primary prostate samples (56.2%) [65]. As expected, the PROfound study identified that age of tumour sample (older tumours had less NGS success), site of biopsy (lymph nodes had greater NGS success compared with bone), and type of primary tumour specimen (prostatectomy specimens had greater likelihood of yielding an NGS result compared with core biopsies) were important factors [7,65,66]. 

In circumstances when tumour samples are not available or the quality of the tumour material is inadequate for NGS, and a re-biopsy is challenging, then germline testing could be performed, with most laboratories currently undertaking panel testing of key germline HRR alterations including *BRCA1/2*. However, it must be remembered that germline testing will not identify somatic events that account for a significant proportion of the HRR alterations seen in prostate cancer. 

Circulating tumour DNA (ctDNA) plasma testing is increasingly considered as an alternative to tumour testing, it has the advantage of capturing both somatic and germline alterations and avoids the need to locate adequate tumour samples or re-biopsy; however, ctDNA can have lower sensitivity in terms of detecting homozygous deletions and result in false negative results, especially if testing is undertaken whilst patients are responding to effective systemic agents with low fractions [36,67]. Additionally, issues of clonal haematopoiesis need to be considered carefully in the interpretation of blood liquid biopsy to avoid false positive results. The PROfound study performed ctDNA alongside tumour testing for HRR and demonstrated a concordance rate of 81% for a positive result and a stronger degree of agreement for negative test results (92%) [7,36,65,66]. Further studies are required to optimise timing and methods for detecting HRR alterations, given molecular heterogeneity between patients and potentially across different sites of mCRPC. It is likely that as the use of PARPi moves to earlier in the disease course, testing will likely be undertaken in the context of diagnosis of mHSPC. 

## 9. Combination Therapeutics with PARP Inhibitors

Studies examining treatment combinations with PARPi in prostate cancer are largely focused on potential therapeutic synergies from combining PARP inhibitors with immunotherapy, targeted agents, and radiation. 

### 9.1. PARP Inhibition and Immunotherapy 

DNA damage from PARP inhibition leads to increase in DNA DSBs, upregulation of PD-L1, and activation of stimulator of interferon genes (STING) signalling; these and other potential synergistic mechanisms can be exploited when combining PARPi and immunotherapy [68,69]. A phase II trial of olaparib and durvalumab (anti-PD-L1) in 17 mCRPC patients unselected for HRR alterations after prior treatment with ARPI demonstrated a radiographic or PSA response in 53% (9/17) [70]. The 12-month PFS was 83.3% (95% CI 27.3–94.5) for patients with HRR alterations and 36.4% (95% CI 11.2–62.7) for patients without HRR alterations (*p* = 0.031) [70]. Whilst patients were unselected for HRR gene alterations at trial entry, two thirds of “responders” were subsequently noted to have HRR alterations, raising questions about the contribution of each therapeutic component [70]. 

KEYNOTE-365 was a phase I/II trial combining olaparib and pembrolizumab in 84 molecularly unselected mCRPC patients that reported a PSA responses rate of 9% (7/82) and ORR 8% (2/24) for RECIST evaluable patients [71]. This combination was further explored in the KEYLYNK-010 phase III trial of pembrolizumab with olaparib vs. an ARPI, in 793 mCRPC patients pre-treated with ARPI and docetaxel. This trial was stopped for futility at the time of interim analysis (median follow-up 11.9 months) after failing to meet the primary endpoints of rPFS or OS. Median rPFS was 4.4 months in the combination arm compared to 4.2 months with ARPI (HR 1.02; 95% CI 0.82–1.25, *p* = 0.55), and OS was 15.8 months vs. 14.6 months (HR 0.94; 95% CI 0.77–1.14, *p* = 0.26) [72]. The CheckMate 9KD phase II study evaluated nivolumab combined with either rucaparib, docetaxel, or enzalutamide across different cohorts in mCRPC patients unselected for HRR alterations [73]. In the post-chemotherapy (1–2 lines of prior taxane regimens) cohort (*n* = 88), patients with HRR alterations demonstrated superior ORR at 17.2% (5.8–35.8) compared to 3.4% (0.1–17.8) in the HRR-negative (or not evaluable) cohort [74]. The QUEST (NCT03431350) phase II study of combination niraparib with abiraterone plus anti-PD1 is currently ongoing [74]. Whilst existing immunotherapy and PARPi studies show modest activity, higher responses have been consistently demonstrated in patients with HRR alterations. 

### 9.2. PARP and Other Targeted Therapies

Ataxia telangiectasia and Rad3-related (ATR) kinase is involved in regulating and stabilising DNA during cell replication. ATR inhibition leads to replication stress and DNA DSBs with synergistic antitumoural effects when administered in combination with PARPi, regardless of HRR status [75,76]. The TRAP phase II trial (NCT03787680) is evaluating the concurrent administration of ceralasertib, an ATR inhibitor, with olaparib in patients with mCRPC across two cohorts with (*n* = 35) and without (*n* = 12) *BRCA1/2* and *ATM* alterations [77]. An early analysis in 2022 revealed that the PSA response was 40% (4/10 patients with sufficient follow-up) amongst patients with HRR alterations. This cohort included patients with *BRCA2* (*n* = 5) and ATM (*n* = 7) alterations [78]. An ongoing basket phase II study (NCT03682289) is currently evaluating the combination of ceralasertib and olaparib in multiple tumour types, including prostate cancer; this trial includes a subgroup of 20 patients with *ATM* loss (10 with mCRPC) [79].

PI3K-AKT signalling is implicated in prostate cancer progression. PARP inhibition has been demonstrated to induce AKT activation from downstream effects on ATM and AKT, potentially promoting acquired resistance to PARPi [80]. Ipatasertib, a potent AKT inhibitor, was combined with rucaparib in a phase Ib trial (NCT03840200) in breast, ovarian, and prostate cancer, based on preclinical data suggesting that synergistic effects may overcome PARPi resistance [81,82]. Preliminary results from the study demonstrated modest combined activity in unselected mCRPC patients with PSA 50 response in 50% patients (3/6) with HRR alterations and 25% (4/16) without [82]. 

Cediranib, an oral ATP-competitive tyrosine kinase inhibitor targeting the vascular endothelial growth factor receptor, has been demonstrated to downregulate expression of *BRCA1*, *BRCA2,* and *RAD51* [83]. In a phase II study randomising 90 patients with mCRPC 1:1 to olaparib either alone or in combination with cediranib, the combination resulted in improved median rPFS 8.5 vs. 4.0 months (HR 0.617; 95% CI 0.392–0.969; *p* = 0.359) in patients unselected for HRR alterations [84].

The biomarker analysis showed that 29% (26/90) of patient biopsies demonstrated HRR alterations, with *BRCA2* alteration accounting for 18% (16/90). The rPFS benefit was numerically greater for the HRR-altered population, with a median rPFS 10.6 with combination therapy vs. 3.8 with olaparib alone, although not significant (HR 0.64; 0.272–1.504) [84]. To expand this concept further, the phase I/II study of durvalumab with olaparib and /or cediranib (NCT02484404) including a cohort of HRR unselected mCRPC patients is currently recruiting [85]. 

REPAIR (NCT05425862) is phase I study assessing the combination of pidnarulex (CX-5461), a drug blocking RNA polymerase I transcription, in combination with talazoparib in an attempt to leverage synergistic effects of replication stress and activation of DNA damage response ultimately leading to cancer cell death [86]. Currently recruiting, this study aims to enrol mCRPC patients unselected for HRR alterations. 

### 9.3. PARP and Radioligand Therapy

PARPi prevent the repair of cytotoxic-induced DNA single-strand DNA damage and have long been used to increase efficacy when given in combination with cytotoxic agents such radiation and chemotherapy. The role of PARPi has evolved beyond external beam radiation therapy to radioligand therapies (RLTs) such as radium-223 and lutetium-177[177Lu]Lu-PSMA-617. RLTs induce DNA breaks that are normally repaired by PARP to enable cell survival; this relationship can be exploited by combining RLT and PARPi to enhance DSBs and enhance cancer cell lethality [87]. Three studies of RLT and PARPi are currently recruiting in mCRPC patients unselected for HRR alterations: LuPARP (NCT03874884) and COMRADE (NCT03317392), assessing [177Lu]Lu-PSMA-617 and olaparib and radium-223 and olaparib, respectively [88,89]. NiraRad (NCT03076203,) a phase Ib trial of radium-223 and niraparib enrolled 30 patients across three PARPi dose cohorts to assess the RP2D and safety profile. Whilst toxicity was manageable, responses were modest, with PSA_50_ at 12 weeks in 10/30 patients [90]. (Table 2).

## 10. Conclusions and Future Directions

Despite the development of new therapeutic strategies with survival advantage, mCRPC remains an incurable disease. There is an urgent need to personalize therapies and improve outcomes for patients with mCRPC. PARPi are now an established standard of care treatment in patients with mCRPC with HRR alternations, namely, *BRCA1/2*. More recently, PARPi in conjunction with ARPI have improved rPFS in mCRPC patients with and without HRR alterations, with the largest benefits seen in patients with *BRCA*1/*2* alterations. Two phase III studies are currently ongoing to explore the efficacy of the combination of PARPi and ARPI therapy in mHSPC with underlying HRR alterations. It is hoped that exploiting the role of PARP on AR signalling and DNA repair, as well as the impact of ARPIs on transcription of HRR genes, by concurrently targeting AR and PARP will extend the duration of castrate sensitivity and modify disease trajectory. In HRR-proficient tumours, PARP inhibition is increasingly being used to enhance the therapeutic potential of DNA-damaging therapies.

The prevalence of HRR alterations in prostate cancer is approximately 25%; however, understanding of prognostic and therapeutic implications of different HRR alterations beyond *BRCA1/2* is currently limited due to small patient cohorts. Further research is critically needed to more accurately define biomarkers of response, thereby enabling improved molecular stratification and optimal selection for PARPi single agent and combination strategies. By incorporating molecular subtyping into ongoing and future clinical trials, greater understanding can be gained on the prevalence of different HRR gene alterations, co-alterations, functional assays, and surrogate markers of the HRR-deficient phenotype in prostate cancer. This, coupled with clinical response, PFS, and OS data, could enable more accurate patient selection and optimal treatment sequencing based on prediction of personalised responses to therapeutics available. With an increasing understanding of prostate cancer heterogeneity and differing molecular subtypes, the era of personalised medicine is in view. 

## Figures and Tables

**Figure 1 cancers-14-05922-f001:**
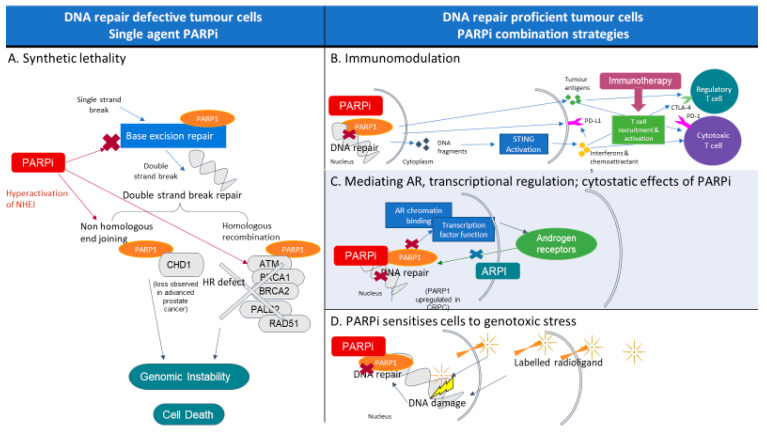
Mechanism of PARPi monotherapy and combination strategies. (HR homologous recombination; STING stimulator of interferon genes; ARPI androgen receptor pathway inhibitors).

**Table 2 cancers-14-05922-t002:** Combination PARP inhibitor trials in metastatic prostate cancer.

Trial Name	Phase of Trial (n = Number of Patients)	Treatment Arms	Patient Population	HRR Biomarker Inclusion	Key Endpoints	Key Efficacy Results
NCT01972217 [46]	Phase II (n = 142)	Olaparib plus abiraterone vs. placebo and abiraterone.	mCRPC following prior docetaxel.	Unselected.	PE: rPFS. SE: PFS 2, OS.	rPFS 13.8 olaparib vs. 8.2 mo placebo (HR 0.65; 95% CI 0.44–0.97; *p* = 0.034). Median PFS2 23.3 (95% CI 17.4—not reached) olaparib vs. 18.5 mo (16.1–23.8) placebo (HR 0.79; 95% CI 0.51–1.21; *p* = 0.28). Median OS 22.7 (95% CI 17.4–29.4) olaparib vs. 20.9 mo (17.6–26.3) placebo (HR 0.91; 95% CI 0.60–1.38; *p* = 0.66).
TALAPRO-2 [49]	Phase III (n = 1095)	Talazoparib plus enzalutamide vs. placebo and enzalutamide.	mCRPC, first-line (ARPI allowed in the castration sensitive setting).	Cohort 1: HRR unselected. Cohort 2: selected for HRR gene alterations.	PE: rPFS. SE: OS.	Prelim: rPFS benefit with talazoparib combination exceeded prespecified HR 0.696.
PROpel [47]	Phase III (n = 796)	Olapaib plus abiraterone and prednisolone vs. placebo abiraterone and prednisolone.	mCRPC, first-line (study allowed for prior treatment with docetaxel in the first-line hormone sensitive setting).	Unselected	PE: rPFS. SE: OS.	Prelim: rPFS 24.8 olaparib and abiraterone vs. 16.6 mo (HR 0.66, 0.95% CI 0.54–0.81; *p* < 0.0001). HRRm cohort: median rPFS 28.8 olaparib and abiraterone vs. 13.8 mo (HR 0.45; 95% CI 0.31–0.65). Non-HRRm cohort: median rPFS 27.6 olaparib and abiraterone vs. 19.1 mo (HR 0.72; 95% CI 0.56–0.93). Cohort with *BRCA* alterations: median rPFS NR olaparib and abiraterone vs. 8.4 mo (HR 0.18; 95% CI 0.09–0.34). Cohort without *BRCA* alterations: median rPFS 27.6 olaparib and abiraterone vs. 16.6 mo (HR 0.72; 95% CI 0.58–0.90). OS data immature.
CASPAR [51]	Phase III (n = 984)	Rucaparib plus enzalutamide vs. placebo and enzalutamide.	mCRPC treatment-naïve.	Unselected.	PE: rPFS, OS. SE: rPFS in *BRCA1/*2 and *PALB2*.	*Ongoing.*
MAGNITUDE [48]	Phase III (n = 670)	Niraparib plus abiraterone vs. placebo and abiraterone.	mCRPC treatment-naïve (allowed for patients who had received less than four months prior abiraterone therapy).	Cohort without HRR alterations and HRR altered cohort (BM+): *ATM, BRCA1, BRCA2, BRIP1, CDK12, CHEK2, FANCA, HDAC2, PALB2.*	PE: rPFS in *BRCA1/2* alterations and BM+ cohort. SE: OS.	Prelim: rPFS in *BRCA1/2*: 16.6 vs. 10.9 mo (HR 0.53 95%CI 0.36–0.79); *p* = 0.0014). rPFS in BM+ cohort 16.5 vs. 13.7 mo (HR 0.73 95% CI 0.56–0.96); *p* = 0.0217).
TALAPRO-3 [52]	Phase III (n = 550)	Talazoparib plus enzalutamide vs. placebo and enzalutamide.	mHSPC.	Selected for alterations in *ATM, ATR, BRCA1, BRCA2, CDK12, CHEK2, FANCA, MLH1, MRE11A, NBN, PALB2, RAD51C.*	PE: rPFS. SE: OS.	*Ongoing.*
AMPLITUDE [53]	Phase III (n = 788)	Niraparib plus abiraterone vs. placebo and abiraterone.	mHSPC (allowed less than 6 months of ADT prior to randomisation and <3 y total).	Selected for HRR gene alteration.	PE: rPFS. SE: OS.	*Ongoing.*
TRAP [78]	Phase II (n = 47)	Ceralasertib (AZD 6738; ATR inhibitor) plus olaparib.	Multiple solid organ malignancies, includes mCRPC cohort with previous ARPI (n = 10).	Cohort 1 with and Cohort 2 without HRR alterations.	PE: Composite response (radiological or PSA_50_) SE: PFS.	Prelim: Cohort 1 (HRR alterations): PSA_50_ 40%.
NCT03840200 [82]	Phase Ib (n = 51)	Ipatasertib (AKT inhibitor) plus rucaparib.	Breast, ovarian, or prostate cancer: including progressive mCRPC with prior ARPI.	Unselected.	Key PE: PSA_50_, dose-limiting toxicities, recommended phase II dose. SE: ORR, rPFS.	Prelim: PSA_50_ 22% all mCRPC patients, 50% in HRR alterations, 25% without HRR alterations. ORR 4% all patients.
NCT02893917 [84]	Phase II (n = 90)	Cediranib plus olaparib (Arm A) or olaparib alone (Arm B).	mCRPC ≥ 1 prior line of therapy.	Unselected.	PE: rPFS. SE: ORR, PSA_50._	Median rPFS 8.5 Arm A vs. 4.0 mo (HR 0.617, 95% CI 0.392–0.969 = 0.0359). ORR 19% Arm A vs. 12%.
NCT02484404 [70]	Phase I/II (n = 384)	Durvalumab plus olaparib and/or cediranib.	mCRPC previously treated with ARPI.	Unselected.	PE: clinical efficacy. SE: ORR, PSA_50._	Radiographic and/or PSA response in 53%. All patients: 12-month rPFS 51.5%. Median rPFS for cohort with HRR alterations: 16.1 mo.
REPAIR [86]	Phase I (n = 48)	Pidnarulex (CX-5461) plus talazoparib.	mCRPC prior taxane, and ARPI.	Unselected.	PE: maximum tolerated dose. SE: PSA_50_, rPFS, ORR, OS.	*Ongoing.*
NCT03810105 [91]	Phase II (n = 5)	Olaparib plus durvalumab.	Castration sensitive biochemically recurrent nonmetastatic prostate cancer.	HRR deleterious mutations.	PE: undetectable PSA.	*Ongoing.*
KEYNOTE-365 [71]	Phase Ib/2 (Cohort A n = 84)	Olaparib with pembrolizumab.	Cohort A: mCRPC post docetaxel, up to 2 prior ARPI.	Unselected.	Key PE: PSA_50_ ORR. SE: OS, rPFS.	Prelim: PSA_50_ 9%. ORR 8%. OS 14 mo.
KEYLYNK-010 [72]	Phase III (n = 793)	Olaparib with pembrolizumab vs. ARPI.	mCRPC progressed with prior ARPI and docetaxel.	Unselected.	PE: rPFS and OS. SE: ORR.	*Stopped for futility in 2022*. Median rPFS 4.4 pembrolizumab and olaparib vs. 4.2 mo ARPI (HR 1.02; 95% CI 0.82–1.25). OS 15.8 pembrolizumab and olaparib vs. 14.6 mo (HR 0.94; 95% CI 0.77–1.14) ORR 17% vs. 6% ARPI (*p* = 0.002).
CheckMate 9KD [73]	Phase II (n = 81 cohort A1)	Nivolumab combined with rucaparib (Arm A), docetaxel (Arm B) or enzalutamide (Arm C).	mCRPC (Arm A1: post 1–2 prior taxane regimes ≤ 2 ARPIs).	Unselected.	PE: ORR, PSA_50_. SE: rPFS, OS.	Cohort A1: ORR 10.3%, all, 17.2% HRR alterations, 3.4% without. PSA_50_ 11.9% all, 18.2% HRR alterations, 5% without. All patients in cohort A1: Median rPFS 4.9 mo. Median OS 13.9 mo.
QUEST [74]	Phase II (n = 136)	Niraparib with abiraterone plus cetrelimab.	mCRPC.	Unselected.	Key PE: ORR, composite response (radiological, CTC, or PSA_50_ response). SE: OS.	*Ongoing.*

PE: primary endpoint(s); SE: key secondary endpoint(s); rPFS: radiological progression-free survival; PFS 2: time to second progression; OS: overall survival; HR: hazard ratio; CI: confidence interval; mo: months; prelim: preliminary results; ARPI: androgen receptor pathway inhibitor; HRR: homologous recombination repair; HRRm: HRR gene alterations; non-HRRm: no HRR gene alterations; NR: not reached; BM+: biomarker-positive (i.e., cohort with HRR alterations); PSA_50_: reduction in PSA level of ≥50%; CTC: circulating tumour cell response.

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
