# Peer review of "Clinical Application of Poly(ADP-ribose) Polymerase (PARP) Inhibitors in Prostate Cancer"

_cancers, 2022, doi:10.3390/cancers14235922_

Round 1

Reviewer 1 Report

The authors provided a comprehensive review of the trials with PARP inhibitors in CRPC patient cohorts, both mono therapy trials and combinatorial trails are presented. The review is thorough and provided intensive information on the current progress of the trial results. In addition, the authors also discussed the various molecular subtyping in the trials using PARP inhibitors. 

The discussion/conclusion could use additional contents in how molecular subtyping and precision medicine approaches can be applied in trials. 

In table 2, key effective results section for NCT0197 2217 (46), no HR,CI and p-value was provided for median PFS2 and median OS. 

Reviewer 2 Report

In the present review titled ‟Clinical Application of Poly (ADP-ribose) polymerase (PARP) Inhibitors in Prostate Cancer”, authors thoroughly summarized the PARP family members, their inhibitors and different therapeutic strategies to treat Stage IV (metastatic) prostate cancer using PARP inhibitors alone or in combination with other agents. The manuscript is well-written and easy to follow. Authors included latest data especially clinical trials is very informative and fine detailed. This review provides an easy and clear understanding of PARP inhibitors and their scope in developing new therapeutics for metastatic castrate resistant prostate cancer (mCRPC).

I recommend this article is suitable for publication. 

Reviewer 3 Report

This is a very important and interesting review summarizing and critically discussing knowledge on clinical application of Poly (ADP-ribose) polymerase (PARP) inhibitors in prostate cancer. The Review is complete and well presented.

I think that it can be accept in present form.
